# Comparative Long-Term Renal Allograft Outcomes of Recurrent Immunoglobulin A with Severe Activity in Kidney Transplant Recipients with and without Rituximab: An Observational Cohort Study

**DOI:** 10.3390/jcm10173939

**Published:** 2021-08-31

**Authors:** Wiwat Chancharoenthana, Asada Leelahavanichkul, Wassawon Ariyanon, Somratai Vadcharavivad, Weerapong Phumratanaprapin

**Affiliations:** 1Tropical Nephrology Research Unit, Department of Clinical Tropical Medicine, Faculty of Tropical Medicine, Mahidol University, Bangkok 10400, Thailand; weerapong.phu@mahidol.ac.th; 2Department of Microbiology, Faculty of Medicine, Chulalongkorn University, Bangkok 10330, Thailand; Asada.L@chula.ac.th; 3Translational Research in Inflammatory and Immunology Research Unit (TRIRU), Department of Microbiology, Faculty of Medicine, Chulalongkorn University, Bangkok 10330, Thailand; 4Cardiometabolic Centre, Department of Medicine, Bangkok Nursing Hospital, Bangkok 10500, Thailand; jowassawon@gmail.com; 5Department of Pharmacy Practice, Faculty of Pharmaceutical Sciences, Chulalongkorn University, Bangkok 10330, Thailand; Somratai.V@chula.ac.th

**Keywords:** immunoglobulin A, kidney transplantation, recurrent glomerulonephritis, rituximab, renal allograft outcomes

## Abstract

Recurrent IgA nephropathy (IgAN) remains an important cause of allograft loss in renal transplantation. Due to the limited efficacy of corticosteroid in the treatment of recurrent glomerulonephritis, rituximab was used in kidney transplant (KT) recipients with severe recurrent IgAN. A retrospective cohort study was conducted between January 2015 and December 2020. Accordingly, there were 64 KT recipients with biopsy-proven recurrent IgAN with similar baseline characteristics that were treated with the conventional standard therapy alone (controls, *n* = 43) or together with rituximab (cases, *n* = 21). All of the recipients had glomerular endocapillary hypercellularity and proteinuria (>1 g/d) with creatinine clearance (CrCl) > 30 mL/min/1.73 m^2^ and well-controlled blood pressure using renin–angiotensin–aldosterone blockers. The treatment outcomes were renal allograft survival rate, proteinuria, and post-treatment allograft pathology. During 3.8 years of follow-up, the rituximab-based regimen rapidly decreased proteinuria within 12 months after rituximab administration and maintained renal allograft function—the primary endpoint—for approximately 3 years. There were eight recipients in the case group (38%), and none in the control group reached a complete remission (proteinuria < 250 mg/d) at 12 months after treatment. Notably, renal allograft histopathology from patients with rituximab-based regimen showed the less severe endocapillary hypercellularity despite the remaining strong IgA deposition. In conclusion, adjunctive treatment with rituximab potentially demonstrated favorable outcomes for treatment of recurrent severe IgAN post-KT as demonstrated by proteinuria reduction and renal allograft function in our cohort. Further in-depth mechanistic studies with the longer follow-up periods are recommended.

## 1. Introduction

The incidence of recurrent immunoglobulin A nephropathy (IgAN) post-kidney transplantation (KT) increases in correlation with the duration of transplantation [1,2], and IgAN post-KT is categorized as a poor prognostic factor [3]. Moreover, the worsening renal allograft function is also associated with the characteristics of severe and active histopathology of IgAN such as endocapillary hypercellularity, crescentic formation, and thrombotic microangiopathy [4,5]. Indeed, the first three histopathological lesions of renal allograft loss in KT with recurrent IgAN are segmental glomerulosclerosis (100%), endocapillary hypercellularity (91.6%), and mesangial hypercellularity (83.3%) [5]. Unfortunately, there is no effective treatment for recurrent IgAN post-KT [6,7] because of the limited effectiveness of renin–angiotensin–aldosterone (RAS) blockade in this condition. Although RAS blockade attenuates proteinuria and possibly preserves kidney function in recurrent IgAN in the early studies [8], the recent studies mention a reduced glomerular filtration rate (GFR) and decreased haematocrit shortly after RAS blockade administration, especially in the recipients with a GFR < 30 mL/min/1.73 m^2^ [9]. The treatment of KT recipients with recurrent IgAN is still based on empirical strategies and remains the main area of uncertainty for transplant nephrologists. In addition, despite RAS blockade treatment, 30–50% of recipients with recurrent IgAN finally lose their renal allograft [5,10,11]. Moreover, methylprednisolone pulse therapy or increased doses of oral steroids in recurrent IgAN post-KT demonstrate the unfavorable outcomes [11] and enhanced steroid side effects, including metabolic disturbances, steroid-induced cataracts and overt immune suppression, in KT recipients.

Rituximab, a chimeric monoclonal anti-CD20, is beneficial in several recurrent glomerular diseases [12,13,14,15,16,17], including in IgAN [12,18,19], through the attenuation of B-cell function. Most rituximab dosing strategies for either de novo or recurrent glomerular diseases have mirrored the standard weekly 375 mg/m^2^ for four doses for non-Hodgkin lymphoma. However, studies have shown no benefit in rituximab administration against recurrent and primary IgAN [20,21]. This is perhaps because a single rituximab administration may not be enough to maintain the effect against B cell networking. Indeed, rituximab possibly reduces autoantibodies against polymeric galactose-deficient IgA1 (Gd-IgA1) [22] and improves renal allograft function in recurrent IgAN [23]. Nevertheless, the risk of infection may increase after rituximab treatment, and a balance between risk and benefit post-KT is necessary. Recognizing this, we have used rituximab monthly 375 mg/m^2^/dose for four doses to treat recurrent post-KT IgAN since 2015. In this study, we sought to investigate the long-term outcomes and safety of the rituximab regimen for endocapillary hypercellularity recurrent IgAN post-KT as an additional treatment to the conventional standard care of RAS blockade plus corticosteroids.

## 2. Materials and Methods

### 2.1. Study Designs and Participants 

A retrospective cohort study between January 2015 and December 2020 was conducted in accordance with STROBE guideline and regulations for humans. The experimental protocol was approved by the Ethics Committee of the Faculty of Tropical Medicine, Mahidol University (MUTM 2021-018-01) and registered in the Thai Clinical Trials Registry (TCTR20210427009). Written informed consent was obtained from participants. The clinical and research activities were consistent with the Principles of the Declaration of Istanbul, as outlined in the Declaration of Istanbul on Organ Trafficking and Transplant Tourism. The conventional standard treatment in both groups consisted of angiotensin-converting enzyme inhibitors (ACEIs) and/or angiotensin receptor blockers (ARBs) to achieve a blood pressure goal of <130/80 mmHg. The inclusion criteria were patients aged between 18 and 70 years with a baseline proteinuria >1 g/24 h, while on stable doses of RAS blockage for at least 2 months and an available renal allograft pathology of recurrent IgAN with endocapillary proliferation. The diagnosis of recurrent IgAN after KT was based on renal allograft biopsies with the protocol biopsy or clinical indications. Only recipients with newly diagnosed recurrent IgAN were eligible for the enrollment. The exclusion criteria were recipients with (i) IgAN in combination with other glomerulonephritis, (ii) IgAN without endocapillary hypercellularity, (iii) previous treatment with rituximab for any other conditions, (iv) contraindications for rituximab such as cirrhosis, active liver disease or hepatitis B and/or C, and active systemic infection or a history of serious infection within 1 month before administration, and (v) missing clinical and laboratory data or referring to another center. There was no corticosteroid dose alteration during the study period, and the dual RAS blockade was prescribed in 5% of the recipients. Meanwhile, mycophenolate mofetil, tacrolimus, and cyclosporin dose adjustments were prescribed following center’s protocol as described below. The estimation of sample size for comparing two means was calculated based on the graft survival rate of recurrent IgAN from Moroni et al. [5]. As a result, 17 participants should be enrolled for each group. No participants from the study group or the control group were lost during the study period.

To assess the histopathological effects of rituximab, the most recent renal allograft biopsy conducted prior to rituximab treatment was compared with the subsequent biopsy by a pathologist in a blinded fashion. In addition to strongly positive IgA staining in part of the immunofluorescence (IF), the light microscopic (LM) characteristics, including mesangial expansion, endocapillary hypercellularity, glomerulosclerosis, and interstitial fibrosis (on silver- and trichrome-stained tissues), were semi-quantitatively scored using MEST-C classification [24] as follows: (i) mesangial hypercellularity (M) was scored from 0 to 1 (0 ≤ 50% of glomeruli showing mesangial hypercellularity, 1 ≥ 50% of glomeruli showing mesangial hypercellularity; (ii) endocapillary hypercellularity (E) was scored from 0 to 1 (0 = no endocapillary hypercellularity, 1 = any glomeruli showing endocapillary hypercellularity); (iii) segmental glomerulosclerosis (S) was scored from 0 to 1 (0 = absent, 1 = present at any glomeruli); (iv) tubular atrophy/interstitial fibrosis (T) was scored from 0 to 3 (0 = 0–25% of cortical area, 1 = 26–50% of cortical area, 2 ≥ 50% of cortical area); and (v) cellular or fibrocellular crescent (C) was scored from 0 to 2 (0 = absent, 1 = 0–25% of glomeruli, 2 ≥ 25% of glomeruli). Electron microscopy (EM) showed electron-dense deposits in the mesangial area without acute rejection in the renal pathology of any of the eligible recipients. Biopsy-proven acute rejection (BPAR) was diagnosed either a “for cause” or a “surveillance” biopsy. BPAR was defined as (i) T-cell-mediated rejection (TMCR) if the histology met the Banff 1997 definition of borderline rejection or higher (i.e., Banff lesion score i ≥ 1 and t ≥ 1) or (ii) antibody-mediated rejection (ABMR) if it met the Banff 2013 criteria for ABMR, including C4d-negative ABMR [25,26].

### 2.2. Immunosuppressive Regimens of Renal Transplantation

Basiliximab induction with maintenance regimens of (i) low-dose corticosteroids (tapered to 5 mg daily after 6 months post-KT), (ii) mycophenolate mofetil (720 mg twice daily of Myfortic^®^ or 1000 mg twice daily of Cellcept^®^), and (iii) either tacrolimus or cyclosporin were used. The target tacrolimus trough level was 8–10 ng/mL for the first 6 months and reduced to 5–8 ng/mL thereafter. Meanwhile, target C2 cyclosporin levels were set to 1000–1200 mg/dL and 800–1000 mg/dL for the first 0–2 and 3–6 months, respectively, and then gradually decreased based on renal allograft function.

### 2.3. Treatment of Recurrent IgAN with Endocapillary Hypercellularity and Cellular Crescent

All the participants in the present study had a 24 h urine protein > 1 g/d. The indication for rituximab administration in this study was recurrent IgAN with endocapillary hypercellularity with or without cellular crescent, based on previous data from our group [23]. Rituximab 375 mg/m^2^/dose was administered monthly for consecutive 4 months with an infection prophylaxis comprising ivermectin (1200 mg/d for 3 days), double-strength trimethoprim–sulfamethoxazole (TMP/SMX) and acyclovir (400 mg/d). To minimize reactions to the rituximab infusions, acetaminophen (1000 mg) was administered orally and diphenhydramine hydrochloride (50 mg) with methylprednisolone (100 mg) intravenously at least 15 min before each infusion. In case of cellular crescent histopathology, 1000 mg/day of pulse methylprednisolone was prescribed for 3 days. 

### 2.4. Outcomes of Interest

The primary outcome of interest was the renal allograft survival (a functioning graft at the last follow-up), which was defined as time elapsing between transplantation and graft loss, either as patient death with a functioning graft or graft failure. Graft failure was defined as the need for permanent dialysis, graft removal, or retransplantation. Patients who were alive with a functioning graft were censored at the date of the last follow-up. Meanwhile, the secondary outcomes of interest were (i) 24 h CrCl at baseline and month 12 of the study, (ii) alterations of 24 h proteinuria between baseline and month 12 of the study, (iii) the episode of biopsy-proven allograft rejection (BPAR) following the treatments, and (iv) treatment complications (i.e., infusion-related reactions, hypogammaglobulinemia, and infections). The response to treatment was qualified as follows: complete remission (CR) and partial remission (PR) indicating a reduction in urine protein to <250 mg/d and at least a 50% reduction from the pretreatment level, respectively.

### 2.5. Sample Analysis

Clinical data of urine and serum were collected with the written informed consent of the participants. Protein and creatinine were measured by colorimetry on an Olympus AU600 autoanalyzer (GmbH, Hamburg, Germany) and nephelometry using Beckman array (Beckman Instruments, Inc., Fullerton, CA, USA), respectively. Because repopulation of the CD19 B cells started 6 months after rituximab administration [23], flow cytometry analysis of CD19 was performed 6–12 months as our practice, but it was not mandatory.

### 2.6. Statistical Analysis

The normality of the data and the homogeneity of the variances were tested using the Shapiro-Wilk and Levene’s tests. Descriptive statistical analyses with range values, including mean (SD) and median, were undertaken. The two-tailed unpaired Student’s *t* test and Mann-Whitney *U* test were used to compare the normally distributed variables and skewed data, respectively, of the 2 groups. Furthermore, the χ^2^ test and Fischer exact test were used for the categorical variables and skewed data, respectively. Two-tailed paired Student’s *t* tests were also used for comparisons before and after each group’s treatment. Correlations were calculated using Fisher’s *r*-to-*z* test. *p* < 0.05 was considered statistically significant. Data analysis was performed using the PASW 18.0.0 statistical software package (SPSS Inc., Chicago, IL, USA) and GraphPad Prism 9.1.0 software (GraphPad Software, Inc., La Jolla, CA, USA).

## 3. Results

A total of 134 patients with recurrent IgAN were diagnosed by either a “for cause” or a “surveillance” biopsy, but only 64 patients with recurrent IgAN with endocapillary proliferation were included in this study. As such, 21 participants received rituximab as an adjunctive therapy to the standard treatment (case), and 43 patients were not administered rituximab (control) (Figure 1). Because of the nonavailability of standard treatment for recurrent IgA after KT, rituximab was administered after a thorough discussion between the physicians and the patients about the potential risks and benefits. The demographic characteristics of all participants (recurrent IgAN post-KT), patients with standard treatment plus rituximab (case) and conventional treatment alone (control) were demonstrated in Table 1 with the similarity between case and control group. The living related KT was predominantly performed in this cohort. The mean age of patients at transplantation was 54 ± 11 years with the median duration from KT to diagnosis of recurrent IgAN being 5.4 years (interquartile range, 3.2–13.5 years). Notably, all recipients experienced at least one episode of proteinuria (UPCR > 0.5 mg/g) during the follow-up period, and most of the participants (87%) had a clinical diagnosis of nephrotic syndrome with a mean of maximum proteinuria 3.87 g/d (range, 1.3–9.0 g/d) at the baseline. Although pulse methylprednisolone 1000 mg/day for three days was commonly prescribed in patients with cellular crescent IgAN, only three patients (from a total of eight patients) in the rituximab with standard treatment (case group) and five patients (from total of six patients) in the conventional treatment alone (control group) received pulse methylprednisolone during the observation. All the participants received at least ACEIs or ARBs to achieve their blood pressure goals, and there was no preemptive transplantation as well as ABO-incompatible KT in this cohort. All participants have completed the study periods without either mortality or loss to the follow-up. Regarding immunosuppression, only tacrolimus and cyclosporin were adjusted following trough and C2 levels, respectively. There was no dose adjustment of both prednisolone and mycophenolate mofetil during the study periods.

### 3.1. Rituximab Attenuated Recurrent IgA Nephropathy with Endocapillary Hypercellularity Pattern in Post-KT

During the median 3.8 years (interquartile 3.1, 4.4 years) of follow-up, 24 h urine protein excretion that was lower than the baseline was more frequently detected in patients with rituximab plus standard treatment compared with standard therapy alone at 12 months after the intervention (Figure 2a). Although proteinuria continuously decreased, only 8 cases (38%) of the recipients in rituximab group achieved a complete remission (CR) at 12 months (mean proteinuria = 0.2 ± 0.1 g/24 h). There was no improvement in 24 h urine protein excretion in the conventional standard treatment group during the study period. In addition, renal allograft function (assessed by 24 h CrCl) demonstrated a significant improvement as early as 18 months postrituximab compared with the control group (Figure 2b).

Additionally, a total of eight cases (38%) of the rituximab group had a CR at 12 months of treatment, and the CR still persisted at 36 months in six cases (29%) of these recipients. There was also a significant difference between the rituximab with conventional standard treatment versus the standard treatment alone in terms of partial remission (PR), *p* < 0.005, at 24 months after therapy leading to a better renal allograft survival in rituximab group (Figure 3a). Moreover, the standard treatment alone had a higher proportion of participants with longer duration of proteinuria during follow-up. Only three cases (14%) in the rituximab with the standard treatment group compared, with 32 cases (74%) in the conventional treatment alone never achieving at least one episode of PR. Of note, 7 cases of these 32 recipients (22%) in the standard treatment alone showed a rapid deterioration in renal allograft function and required renal replacement therapy within 24 months. The median CR duration in the rituximab group within the 3 year follow-up period was 27.5 months (range: 14.2 to 36 months). More importantly, at the end of study, three cases (14%) of the rituximab group had allograft loss, as compared to 22 cases (51%) of the conventional standard treatment alone (Figure 3b), which demonstrated the better graft survival in rituximab group by Kaplan-Meier estimation (*p* = 0.002). Notably, the only cause of allograft loss from both groups was the progression of the recurrent disease.

With respect to renal allograft histopathology, all of the recipients had diffuse endocapillary hypercellularity (E score = 1 according to the MEST-C criteria in the updated Oxford classification of IgA nephropathy) at baseline. Cellular or fibrocellular crescentic formation was presented in 14 of the 64 recipients (six cases in the standard treatment alone and eight cases in the rituximab group). Moreover, it should be emphasized that participants from both groups had minor chronicity of pathological background defined by tubular atrophy/interstitial fibrosis (Figure 4a). Histopathological improvement following MEST-C criteria in the updated Oxford classification of IgA nephropathy was found in the vast majority of the recipients but was significantly better in the rituximab group (Figure 4a). Of particular note, although only sixty percent of recipients received a follow-up renal allograft biopsy, and the renal allograft histopathology following rituximab administration was less severe in both endocapillary and mesangial hypercellularity, despite the remaining strong IgA deposition. Unlike rituximab with the conventional standard treatment group, the standard treatment alone showed more progression of interstitial fibrosis/tubular atrophy at the 12 month follow-up renal allograft biopsy. 

The recipients in the rituximab group (*n* = 18) had complete peripheral B-cell depletion within the first month of rituximab administration. Moreover, CD19 B cells were again detectable in the blood after the last dose of rituximab administration at a mean time of 8.9 months. In the recipients who achieved a CR within 12 months, there was a significantly longer duration before the repopulation of CD19 B cells compared with the recipients who did not achieve CR (*p* = 0.04) (Figure 4b). Thus, the number of CD19 cells in blood could possibly be useful to predict the rituximab responsiveness. 

### 3.2. Episodes of Allograft Rejection following the Treatments

There was no report of biopsy-proven allograft rejection (BPAR), including acute cellular rejection, acute antibody-mediated rejection, or combined acute cellular and antibody-mediated rejection, in the recipients of both groups during the study period. Interestingly, six out of nine recipients in the rituximab with conventional standard treatment demonstrated positive C4d deposits (but no peritubular capillaritis) with a decrease in the activity score in 12 month allograft pathology (mean C4d score from 1.8 to 0.3; *p* < 0.05). On the other hand, only two of fifteen recipients in the standard treatment alone had a decrease in the activity score at 12 month pathology (mean C4d score from 1.6 to 1.2; *p* = 0.62). However, due to the practice protocol, we did not perform donor-specific antibodies unless the recipients had been suspected for acute/active antibody-mediated rejection.

### 3.3. Adverse Events

The adverse events were analyzed on an intention-to-treat basis through to the last day of the study. There was no reported case of either infusion reaction or serious adverse events defined as >1 episode of treatment-related leukopenia and severe systemic infection related to rituximab.

## 4. Discussion

This is a first longitudinal study of rituximab administration in kidney transplant (KT) recipients with recurrent endocapillary hypercellularity IgAN. We demonstrated that the adjunctive treatment of four doses rituximab plus the conventional standard renin–angiotensin–aldosterone (RAS) blockade increased the proportion of patients with a clinically complete remission (CR), reduced proteinuria, and maintained renal allograft function after 12 months of the initial rituximab dose in comparison with RAS blockage alone.

Recurrent IgAN is a leading cause of recurrent glomerular disease after kidney transplantation, affecting almost 50% of recipients after 5 years post-KT [27]. Interestingly, 30% of recipients with recurrent IgAN have a progressive loss of renal allograft function and reach allograft failure stage prematurely [4,5]. There is currently no evidence to support any specific therapeutic regimens for the recurrence of IgAN following KT although the Kidney Disease: Improving Global Outcomes (KDIGO) Transplant Work Group has suggested that treatment with angiotensin-converting enzyme inhibitors (ACEIs) or angiotensin receptor blockers (ARBs) may reduce the rate of decline in renal allograft function [6]. IgA1 with some *O*-glycans deficient in galactose (galactose-deficient IgA1; Gd-IgA1)-forming an immune complex due to IgG and/or IgA autoantibodies is currently thought to be a key pathogenesis of primary IgAN [22,28] and recurrent IgAN [29]. As such, observational retrospective analyses have demonstrated a lower rate of recurrent IgAN in recipients with antithymocyte globulin induction, which may be due to the eradication of B cells [30]. Additionally, the database of the Australia and New Zealand Dialysis and Treatment Registry (ANZDATA) and the United Network for Organ Sharing (UNOS)/Organ Procurement and Transplantation Network (OPTN) shows a higher incidence of IgAN in recipients with a steroid withdrawal regimen, perhaps due to an increase in the immune complex [31,32]. 

At present, the treatment of recurrent IgAN should aim to reduce proteinuria and decrease inflammation as suggested by the KDIGO transplant recommendation [6]. Of note, all of the participants in this cohort received standard supportive therapy, in particular the use of ACEIs or ARBs, for controlling either blood pressure or proteinuria. Here, rituximab-attenuated renal allograft deterioration through decreased proteinuria with a gradually improved allograft function started after 9 months of the treatment (Figure 2) that resulted in more favorable outcome (5-year renal allograft survival) when compared with the control group. Although the positive effect on proteinuria was no longer beyond 36 months (Figure 2a), the complete remission of proteinuria may improve the prognosis [33]. In addition, rituximab has a direct target against B cells, and the decrease immune cells in the renal allograft may reduce the risk of a consequential inflammatory progression to the clinically significant renal allograft rejection [34]. Furthermore, B cells play a central role in the immunopathogenesis of glomerulonephritis, including IgAN [19,35]. B-cell activating factor (BAFF) and a proliferation-inducing ligand (APRIL) are proteins of the tumor necrosis factor superfamily that interact through three receptors, including (BAFF receptor (BAFF-R), transmembrane activator, and cyclophilin ligand interactor), to promote B-cell survival and B-cell maturation into memory B cells and plasma cells [36]. High expression of BAFF and APRIL involves the pathogenesis of IgAN for the increased production for antibodies against the glycan groups of IgA [37]. These data suggest a possible role of B cells in patients with recurrent IgAN after kidney transplantation. 

Interestingly, the repopulation of CD19 B cells after rituximab administration in the responsive recipients (with CR) was longer than the recipients without CR implying a possible impact of B cells on the activity of IgAN. Notably, CD19 alone may not be sufficient to induce the maturation of all B-cell repertoires [38]. Other B-cell proliferation parameters, such as miR-374b, might be the additional factors that are associated with recurrent IgAN post-KT [39]. In addition, rituximab may be protective against antibody-mediated allograft rejection [40]. Although we could not find the beneficial effect of rituximab toward allograft rejection in the present study, six out of nine (66%) recipients in the rituximab group with positive C4d deposits, but who were not shown peritubular capillaritis, had a significant decrease in C4d activity at 12 months of rituximab treatment. In the future, more detailed analysis is warranted on the potential relationship between donor specific antibodies, mean fluorescence intensity, C4d activity, and clinical response of rituximab in term of recurrent IgAN. Nevertheless, rituximab, with a less frequent administration in post-KT recurrent IgAN, has failed to reduce serum Gd-IgA1 in primary IgAN [21], perhaps due to the differences in several aspects between post-KT recurrent IgAN and primary IgAN, including natural history, stage, and severity of IgAN and the prescribed rituximab doses. Further studies are clearly necessary. Despite the major concerns on infection after rituximab treatment, the infectious complications in rituximab group in our study were not different from the conventional standard treatment group, perhaps due to the strategic prevention using ivermectin, TMP/SMX, and acyclovir. Rituximab also influences T cells [41] and may induce T-cell-dependent viral infections such as varicella zoster virus (VZV), cytomegalovirus (CMV), and chronic hepatitis B. Thus, the close monitoring of CMV and HBV DNA during the co-administration of rituximab with other T-cell-mediated immunosuppressants such as CNIs and MMF may be necessary [42]. Further studies in this area are needed.

The critical and unique value of the study is that we demonstrated the potential efficacy and safety use of rituximab in recurrent IgA nephropathy with a high disease severity, mostly with the endocapillary and mesangial hypercellularity in histopathological. Additionally, rituximab was also effective in an early stage of IgAN with fewer chronicity indexes as defined by interstitial fibrosis-tubular atrophy. Nevertheless, the results could not be generalized for all stages and severities of recurrent IgAN post-KT. Second, several molecular aspects of rituximab administration in IgAN, such as the autoantibody against Gd-IgA1 [43], cytoskeleton stabilization [44], and T-cell modulation [41], were not explored. Third, due to the limitation of retrospective study, confounding treatment effects may exist, including the probability of selection bias, particularly in the case selection for treatment with rituximab. However, the utilization of cyclosporine, tacrolimus, azathioprine, mycophenolate mofetil, sirolimus, or prednisone was independently associated with renal allograft loss that did not prevent renal allograft loss from the post-KT recurrent glomerulonephritis, including IgAN [45].

## 5. Conclusions

In conclusion, our results demonstrated the potential efficacy and safety of rituximab in KT recipients with recurrent IgAN, particularly with endocapillary hypercellularity in renal pathology, in the attenuation of proteinuria and improved renal allograft function. Accordingly, rituximab is an interesting candidate adjunctive treatment with RAS blockage against recurrent IgAN after KT.

## Figures and Tables

**Figure 1 jcm-10-03939-f001:**
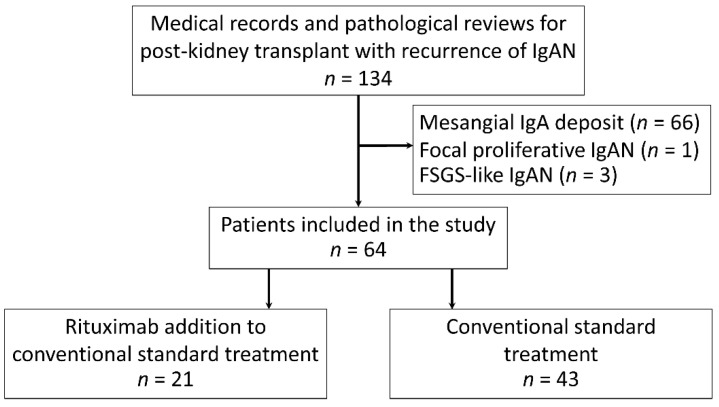
Flowchart of participants included in the study. FSGS: focal segmental glomerulosclerosis; IgA: immunoglobulin A; and IgAN: immunoglobulin A nephropathy.

**Figure 2 jcm-10-03939-f002:**
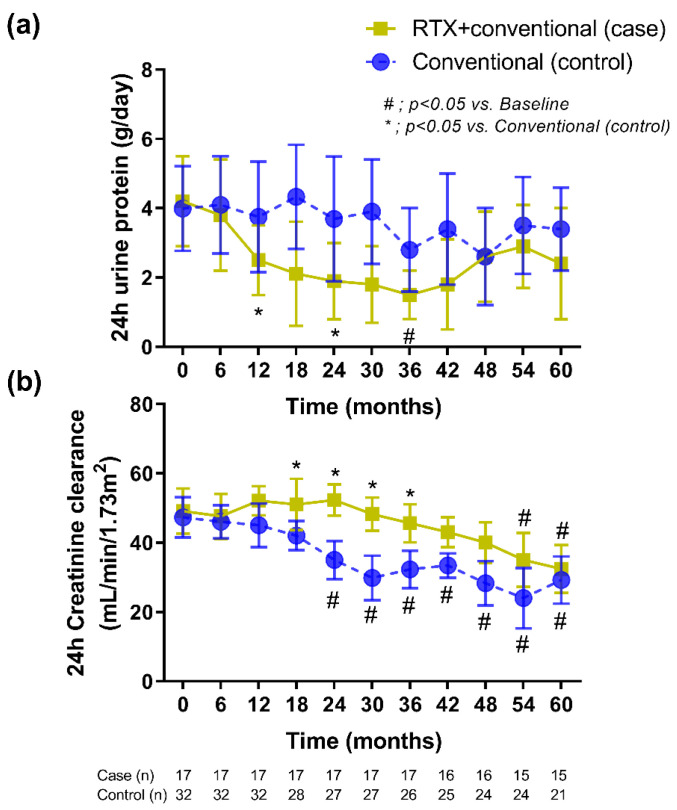
The treatment responses in the kidney transplant recipients with rituximab administration plus the conventional standard treatment versus those with the conventional standard treatment alone as demonstrated by (**a**) 24 h proteinuria and (**b**) 24 h creatinine clearance. The data included only recipients who had a follow-up duration ≥36 months with censoring in whom there was nonfunctioning graft (or required long-term renal replacement therapy). RTX, rituximab.

**Figure 3 jcm-10-03939-f003:**
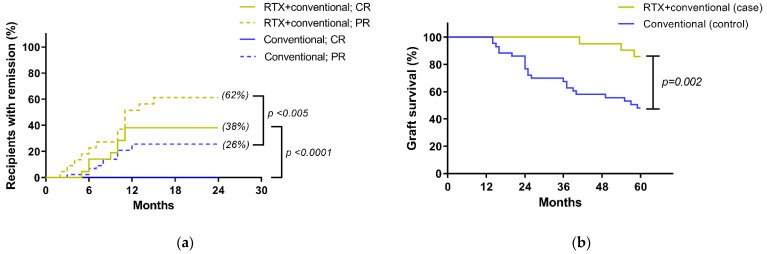
Treatment outcomes: (**a**) the percentages of recipients in each of the 4 subgroups according to the treatment response indicated as complete remission (CR) or partial response (PR) (see Method); (**b**) Kaplan-Meier estimates of renal allograft survival in recipients correspond to treatment group; RTX, rituximab.

**Figure 4 jcm-10-03939-f004:**
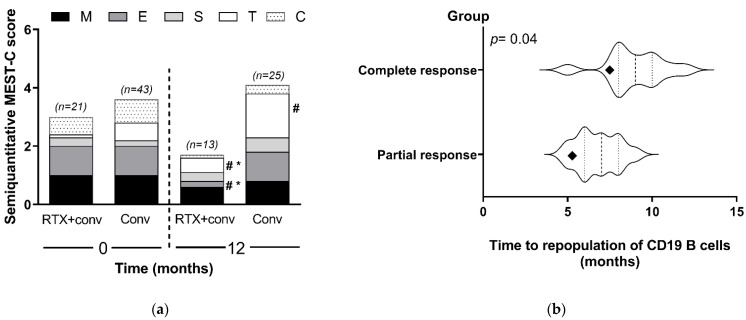
Post-treatment histopathology and immune cells: (**a**) histologic information on renal allograft biopsy at baseline and at 12 months is demonstrated following the MEST-C criteria in the updated Oxford classification of IgA nephropathy (M = mesangial hypercellularity, M0–M1; E = endocapillary hypercellularity, E0–E1; S = segmental glomerulosclerosis, S0–S1; T = tubular atrophy/interstitial fibrosis, T0–T2; and C = cellular or fibro-cellular crescents, C0–C2). (**b**) The duration of CD19 B-cell repopulation after rituximab administration compared between the patients with a complete response (CR) (*n* = 8) and those with a partial response (PR) (*n* = 10) (participants did not perform testing; *n* = 3). (Violin plots show the differences in the average (diamond), median (middle line) and time of repopulation). Conv, conventional standard treatment group; RTX, rituximab. # *p* < 0.05 versus baseline; and * *p* < 0.05 versus conventional.

**Table 1 jcm-10-03939-t001:** Participants’ baseline characteristics.

Variables	All Recipients (*n* = 64)	Rituximab Addition to Conventional Standard Treatment (*n* = 21)	Conventional Standard Treatment (*n* = 43)	*p*-Value
Recipient age, year	53.9 ± 11.2	53.1 ± 9.2	55.3 ± 6.4	0.047
Recipient gender, *n* (% male)	37 (57.8)	13 (61.9)	24 (55.8)	0.65
Donor type, *n* (% living donors)	59 (92.2)	18 (85.7)	41 (95.3)	0.18
Donor age, year	39.4 ± 6.1	38.3 ± 10.6	41.6 ± 11.2	0.81
Donor gender, *n* (% male)	30 (46.9)	10 (47.6)	20 (46.5)	0.93
HLA mismatch	1.3 ± 0.2	1.3 ± 0.9	1.2 ± 1.0	0.62
PRA > 30%, *n* (%)	14 (21.9)	7 (33.3)	7 (16.3)	0.13
Systolic blood pressure (mmHg)	130.0 ± 14.8	130.6 ± 11.4	129.5 ± 12.4	0.70
Diastolic blood pressure (mmHg)	75.8 ± 9.4	77.2 ± 7.8	73.5 ± 4.5	0.003
Body mass index (kg/m^2^)	23 ± 6.2	23 ± 2.2	24 ± 3.2	0.07
Serum creatinine (mg/dL)	2.1 ± 0.8	2.0 ± 1.1	2.1 ± 1.4	0.25
24 h CrCl (mL/min/1.73m^2^)	48.9 ± 12.8	49.1 ± 9.3	48.7 ± 11.1	0.40
Maximum proteinuria (mg/24 h)	4483	4.51	4043	0.65 ^b^
	(1284–8987) ^a^	(1284–8987)	(1341–7734)	
Smoking, *n* (%)	11 (17.2)	4 (19.0)	7 (16.3)	0.79
Diabetes, *n* (%)	36 (56.2)	12 (57.1)	24 (55.8)	0.92
Cardiovascular disease, *n* (%)	14 (21.9)	6 (28.6)	8 (18.6)	0.37
HBV/HCV, *n* (%)	0 (0)/0 (0)	0 (0)/0 (0)	0 (0)/0 (0)	-
CMV status (D+/R+), *n* (%)	64 (100)	21 (100)	43 (100)	-
Previous allograft rejection, *n* (%)	0 (0)	0 (0)	0 (0)	-
Diagnosis: post-transplant, months	64.2 (38.1, 162.4) ^c^	69.5 (39.2, 172.4)	63.7 (36.8, 153.6)	0.21
Cyclosporin A, *n* (%)	2 (3.1)	1 (4.8)	1 (2.3)	0.59
Tacrolimus, *n* (%)	62 (96.9)	20 (95.2)	42 (97.7)	0.59
Mycophenolate mofetil, *n* (%)	64 (100)	21 (100)	43 (100)	-
Corticosteroids, *n* (%)	50 (78.1)	18 (85.7)	32 (74.4)	0.31
Prednisolone dose (mg/d)	3.3 ± 2.1 ^d^	3.2 ± 2.0	3.3 ± 1.6	0.27
ACEIs or ARBs, *n* (%)	64 (100)	21 (100)	43 (100)	-
ACEIs and ARBs, *n* (%)	3 (4.7)	2 (9.5)	1 (2.3)	0.20
Pulse methylprednisolone, *n* (%) ^e^	8 (12.5)	3 (14.3)	5 (11.6)	0.76

^a^ mean (range); ^b^ Mann-Whitney; ^c^ median and IQR; ^d^ mean prednisolone dose calculated from 50 recipients (*n* = 18 and 32 for case and control groups, respectively); ^e^ the total case of cellular crescent was 14 cases (*n* = 8 and 6 for case and control groups, respectively), and 1000 mg/day of pulse methylprednisolone was prescribed for three days. ACEIs, angiotensin-converting enzyme inhibitors; ARBs, angiotensin receptor blockers; CrCl, creatinine clearance; CMV, cytomegalovirus; D+/R+, positive serology for donor/positive serology for recipient; HBV, hepatitis B virus; HCV, hepatitis C virus; HLA, human leucocyte antigen; IQR, interquartile range; and PRA, panel reactive antibody.

## Data Availability

The data are available from the corresponding author upon reasonable request.

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
