# Peer review of "Comparative Long-Term Renal Allograft Outcomes of Recurrent Immunoglobulin A with Severe Activity in Kidney Transplant Recipients with and without Rituximab: An Observational Cohort Study"

_jcm, 2021, doi:10.3390/jcm10173939_

Round 1

Reviewer 1 Report

Chancharoenthana et al. reported the article of jcm-1279723, entitled "Comparative long-term renal allograft outcomes of severely active recurrent immunoglobulin A in kidney transplant recipients treating with and without rituximab: an observational cohort study". This is a clinically relevant issue regarding recurrent glomerulonephritis in kidney transplant (KTx), especially in Asia. As authors mentioned in this article, no specific treatment for recurrent IgAN after KTx is there so far. Rituximab (Rit) has been recently available for many diseases in the nephrology field, and nephrologists have had plenty of experience; thus, Rit for recurrent IgAN will become a standard of care. This information would be a better consideration for the management of recurrent IgAN. I have some questions before the final judge. Please see below; 

1. Please describe the prevalence of ABO blood type incompatible KTx. Since ABO-incompatible KTx would get Rit as a desensitized therapy.
2. Authors should declare the whole KTs recipients in the study period, then authors describe the percentage of participates. Furthermore, how did the authors deal with the recipients of recurrent IgAN without endocapillary hypercellularity? We would like to know the study flow chart first.
3. How prevalence crescentic region did the recipients have in both groups? If the Rit group had a high prevalence of crescentic region, a high dose of steroid pulse therapy underwent. If yes, the steroid instead of Rit might be effective. The authors should describe the prevalence of steroid pulse therapy in Table 1. 
4. Authors should describe the cause of graft loss. This study focused on the efficacy of Rit for IgAN. If the cause of graft loss were the majority of rejection, the relationship between Rit and recurrent IgAN would not disappear. 
5. please clarify the definition of graft loss. Death censored? Of Death uncensored? 
6. Authors should describe the indication of Rit for the treatment of recurrent IgAN.
7. The risk factors for graft loss, including diabetes, BMI, smoking, episode of rejection, history of CVD and CMV, etc.… should be described in Table 1. 
8. Due to the retrospective design, it is too strong to conclude as “…..rituximab was beneficial……” in conclusion. Retrospective cohort studies never conclude casualty. Please replace it with another phrase.  There was just association. 
9. Please describe the ethical issue regarding the Istanbul declaration. I’m so curious about the relationship of living donors. They were so young, and fewer mismatches in donors. Were they little bros./sisters? Or wives? 

10. Authors should describe the numbers of HCV/HBV in the standard group. 

Author Response

Response to the reviewers 

Comments and Suggestions for Authors

Chancharoenthana et al. reported the article of jcm-1279723, entitled "Comparative long-term renal allograft outcomes of severely active recurrent immunoglobulin A in kidney transplant recipients treating with and without rituximab: an observational cohort study". This is a clinically relevant issue regarding recurrent glomerulonephritis in kidney transplant (KTx), especially in Asia. As authors mentioned in this article, no specific treatment for recurrent IgAN after KTx is there so far. Rituximab (Rit) has been recently available for many diseases in the nephrology field, and nephrologists have had plenty of experience; thus, Rit for recurrent IgAN will become a standard of care. This information would be a better consideration for the management of recurrent IgAN. I have some questions before the final judge. Please see below; 

  1. Please describe the prevalence of ABO blood type incompatible KTx. Since ABO-incompatible KTx would get Rit as a desensitized therapy.

Answer: Because only one center has an experience for ABO-incompatible KT in Thailand so that the prevalence of ABO-incompatible KT is very low. Thus, there was no ABO-incompatible KT in this study. We have added the detail in the Results.

  1. Authors should declare the whole KTs recipients in the study period, then authors describe the percentage of participates. Furthermore, how did the authors deal with the recipients of recurrent IgAN without endocapillary hypercellularity? We would like to know the study flow chart first.

Answer: We thank the reviewer for the comment. However, because the target population in the study was the KT recipients who has developed recurrent IgAN in the duration between January 2015 and December 2020 with no restriction to post-KT duration, so the data of whole KT recipients data may not reflect the true prevalence of recurrent IgAN. In addition, some cases of recurrent IgAN were referred from another center. According to our practice, those recipients of recurrent IgAN without endocapillary hypercellularity, we treated as same as conventional post-KT recipient care without specific treatments. We have added the study flow chart in the Results

  1. How prevalence crescentic region did the recipients have in both groups? If the Rit group had a high prevalence of crescentic region, a high dose of steroid pulse therapy underwent. If yes, the steroid instead of Rit might be effective. The authors should describe the prevalence of steroid pulse therapy in Table 1. 

Answer: Cellular crescentic formation was presented in 14 of the 64 recipients (6 cases in the conventional standard treatment group and 8 cases in the rituximab with conventional standard treatment group). But only 5 cases in the conventional standard treatment group and 3 cases in the rituximab with conventional standard treatment group were treated with pulse MP. We have added the detail in the Table 1.

  1. Authors should describe the cause of graft loss. This study focused on the efficacy of Rit for IgAN. If the cause of graft loss were the majority of rejection, the relationship between Rit and recurrent IgAN would not disappear. 

Answer: We thank the reviewer for the comment. We have corrected accordingly.

  1. please clarify the definition of graft loss. Death censored? Of Death uncensored? 

Answer: We thank the reviewer for the comment. We have corrected accordingly.

  1. Authors should describe the indication of Rit for the treatment of recurrent IgAN.

Answer: We thank the reviewer for the comment. We have corrected accordingly.

  1. The risk factors for graft loss, including diabetes, BMI, smoking, episode of rejection, history of CVD and CMV, etc.… should be described in Table 1. 

Answer: We thank the reviewer for the comment. We have corrected accordingly.

  1. Due to the retrospective design, it is too strong to conclude as “…..rituximab was beneficial……” in conclusion. Retrospective cohort studies never conclude casualty. Please replace it with another phrase. There was just association. 

Answer: We thank the reviewer for the comment. We have corrected accordingly.

  1. Please describe the ethical issue regarding the Istanbul declaration. I’m so curious about the relationship of living donors. They were so young, and fewer mismatches in donors. Were they little bros./sisters? Or wives? 

Answer: We have performed both clinical and research activities in consistent with the Principles of the Declaration of Istanbul, as outlined in the Declaration of Istanbul on Organ Trafficking and Transplant Tourism. We have described accordingly.

    Indeed, the donors of living-related KT in Thailand are mainly from spouse (wives > husbands), brothers, and sisters. All of living donor donation will be conducted by local medical review board system under the regulation of the National Organ Donation, Thai Red Cross.

  1. Authors should describe the numbers of HCV/HBV in the standard group

Answer: We thank the reviewer for the comment. We have corrected accordingly.

Reviewer 2 Report

The manuscript and results are indeed interesting and relevant. It needs some English language editing, particularly for the title, the abstract and the first half of the results. 

It is puzzling to me why the authors did not include rejection episodes, cellular, antibody, or combined ACR/AMR in the results. Patients were biopsied, therefore a grading for rejection must be included. Also, the benefit of rituximab must be adjusted based on number of rejection episodes in each group. Or at least, show that there was no difference in rejection episodes and severity, and Donor specific antibodies between the two groups. 

Finally, make sure to present the allograft survival both censored and uncensored by patient's death. 

Author Response

Comments and Suggestions for Authors

The manuscript and results are indeed interesting and relevant. It needs some English language editing, particularly for the title, the abstract and the first half of the results.

Answer: We have edited the English language as the reviewer suggestion accordingly. 

It is puzzling to me why the authors did not include rejection episodes, cellular, antibody, or combined ACR/AMR in the results. Patients were biopsied, therefore a grading for rejection must be included. Also, the benefit of rituximab must be adjusted based on number of rejection episodes in each group. Or at least, show that there was no difference in rejection episodes and severity, and Donor specific antibodies between the two groups. 

Answer: We thank the reviewer for the comment. We have added the information regarding to rejection in the Results and Discussion. However, there was no surveillance DSA levels in our practice unless the recipients have developed episodes of allograft rejection.

Finally, make sure to present the allograft survival both censored and uncensored by patient's death. 

Answer: We thank the reviewer for the comment. There was no mortality case of participants during the study period in both groups. We have added the information accordingly.

Reviewer 3 Report

This is a retrospective study regarding the use of rituximab (4 monthly doses) in 21 renal transplant recipients with recurrent IgA nephropathy compared with a control group of 43 patients.

The authors report reduction of proteinuria in the rituximab group (8 patients with complete remission) and improved creatinine clearance.

The study is very interesting as it reports a very favorable effect of rituximab in recurrent IgA with severe proteinuria (> 4 gr/24h).

The authors should state in the text the probability of bias by indication for these results as the study was not randomized and they do not clearl explain how the have chosen the patients to receive rituximab.

Also there is no data regarding the tubulointersitial findings of the initial pathology reports as well as the possibility of chronic allograft nephropathy (CAN) or chronic antibody mediated rejection (no data on donor specific antibodies, or C3 staining in the biopsies). I appreciate that this is a very complex scenario but there is a possibility that the favorable effect might be due to these reasons (allo-immunity) and not only on the pathogenesis of IgA nephropathy (although here is no c;ear data that rituximab has a favorable effect on CAMR). However, if there in no data I still consider the paper significant but the authors should add these issues in the limitations of the study.

The authors should also clearly state if there was any other alteration in the immunosuppression regimen (especially on MMF doses for both groups).

Author Response

Comments and Suggestions for Authors

This is a retrospective study regarding the use of rituximab (4 monthly doses) in 21 renal transplant recipients with recurrent IgA nephropathy compared with a control group of 43 patients.

The authors report reduction of proteinuria in the rituximab group (8 patients with complete remission) and improved creatinine clearance.

The study is very interesting as it reports a very favorable effect of rituximab in recurrent IgA with severe proteinuria (> 4 gr/24h).

The authors should state in the text the probability of bias by indication for these results as the study was not randomized and they do not clear explain how the have chosen the patients to receive rituximab.

Answer: We thank the reviewer for the comment. We have added the information accordingly.

Also there is no data regarding the tubulointersitial findings of the initial pathology reports as well as the possibility of chronic allograft nephropathy (CAN) or chronic antibody mediated rejection (no data on donor specific antibodies, or C3 staining in the biopsies). I appreciate that this is a very complex scenario but there is a possibility that the favorable effect might be due to these reasons (allo-immunity) and not only on the pathogenesis of IgA nephropathy (although here is no clear data that rituximab has a favorable effect on CAMR). However, if there is no data I still consider the paper significant but the authors should add these issues in the limitations of the study.

Answer: We thank the reviewer for the comment. We have added the information accordingly.

The authors should also clearly state if there was any other alteration in the immunosuppression regimen (especially on MMF doses for both groups).

Answer: We thank the reviewer for the comment. We have added the information accordingly.

Round 2

Reviewer 1 Report

The authors incorporate my suggestions into the revised version. I don't have additional comments.